# Menstrual Phase Affects Coagulation and Hematological Parameters during Central Hypovolemia

**DOI:** 10.3390/jcm9103118

**Published:** 2020-09-27

**Authors:** Nandu Goswami, Bianca Brix, Andreas Roessler, Martin Koestenberger, Gilbert Reibnegger, Gerhard Cvirn

**Affiliations:** 1Physiology Division, Otto Loewi Research Center, Medical University of Graz, 8036 Graz, Austria; bianca.brix@medunigraz.at (B.B.); andreas.roessler@medungiraz.at (A.R.); 2Department of Pediatric Cardiology, Medical University of Graz, 8036 Graz, Austria; martin.koestenberger@medunigraz.at; 3Physiological Chemistry Division, Otto Loewi Research Center, Medical University of Graz, 8036 Graz, Austria; gilbert.reibnegger@medunigraz.at (G.R.); gerhard.cvirn@medunigraz.at (G.C.)

**Keywords:** coagulation, menstrual phase, thrombelastometry, thrombin generation

## Abstract

Background: It has been reported that women have a higher number of heart attacks in the “follicular phase” of the menstrual cycle. We, therefore, tested the hypothesis that women in the follicular phase exhibit higher coagulability. As lower body negative pressure (LBNP) has been used previously to assess coagulation changes in whole blood (WB) samples in men and women, effects of menstrual phase on coagulation was assessed during LBNP. Methods: Seven women, all healthy young participants, with no histories of thrombotic disorders and not on medications, were tested in two phases of the menstrual cycle (early follicular (EF) and mid-luteal (ML)). LBNP was commenced at −10 mmHg and increased by −10 mmHg every 5 min until a maximum of −40 mmHg. Recovery up to 10 min was also monitored. Blood samples were collected at baseline, at end of LBNP, and at end of recovery. Hemostatic profiling included comparing the effects of LBNP on coagulation values in both phases of the menstrual cycle using standard coagulation tests, calibrated automated thrombogram, thrombelastometry, impedance aggregometry, and markers of thrombin formation. Results: LBNP led to coagulation activation determined in both plasma and WB samples. During both phases, coagulation was affected during LBNP, as reflected in their decreased partial thromboplastin time (PTT) and elevated coagulation factor VIII FVIII, F1 + 2, and thrombin-antithrombin (TAT) levels. Additionally, during the ML phase, greater PT [%] and shorter time to peak (ttPeak) values (implying faster maximum thrombin formation) suggest that women in the ML phase are relatively hypercoagulable compared to the early follicular phase. Conclusions: These results suggest that thrombosis occurs more during the midluteal phase, a finding with substantial medical implications.

## 1. Introduction

Lower body negative pressure (LBNP) suction causes a pressure gradient that pulls fluid from the intravascular to the extravascular space in the lower body [1]. Generally, it leads to blood pooling in the legs, reduces venous return, and has been shown to be a non-invasive surrogate to study physiological responses to blood loss/hemorrhage, a leading cause of death in patients with trauma [2,3]. LBNP-induced hemoconcentration elevates plasma protein concentration and blood viscosity. Overall, the increased interactions between coagulation factors and procoagulant factors released from vascular endothelial cells, together with platelet activation, create a procoagulant milieu, which is seen in both sexes [2,4].

LBNP-induced activation of the coagulation cascade has been reported in most of the studies [5,6,7]. Most of these studies have used platelet-poor plasma (PPP), known to contain most of the coagulation factors, but some have used whole blood (WB), which includes platelets and phospholipid-bearing cells; both of which are involved in supporting coagulation [8]. Therefore, in this study, coagulation status was investigated not only in PPP samples (levels of the pro- and anticoagulatory factors, coagulation times, markers of thrombin formation, thrombin generation curves, and fibrinolytic parameters) but also in WB samples (course of clot development and blood cell counts, as well as platelet function). Red blood cell (RBC), hematocrit (Hct), hemoglobin (Hb), and mean corpuscular hemoglobin (MCH) were automatically measured in all blood count determinations.

Furthermore, the hemostatic system of women differs from that of men in various aspects. For example, women, as compared to men, have lower median levels of the anticoagulant protein S (PS), and thrombelastometry (TEM) measurements show an earlier onset and an elevated coagulation velocity [9,10]. We have recently also reported differences in baseline coagulation parameters between men and women and reported how central hypovolemia, induced by LBNP, affects coagulation responses [4]. We observed that, at baseline, women were hypercoagulable compared to men, as evidenced by their shorter ‘Lag times’ and higher thrombin peaks and by shorter “Coagulation times” and ‘Clot formation times’. Moreover, men were more susceptible to LBNP, as reflected in their elevated FVIII levels and decreased ‘Lag times’ following LBNP. LBNP-induced coagulation activation was also not accompanied by endothelial activation. It appears that application of LBNP might be a useful future tool to identify individuals with an elevated risk for thrombosis in subjects with or without history of thrombosis.

It has been reported that women have a higher number of heart attacks in the “follicular phase” of the menstrual cycle [11]. We, therefore, tested the hypothesis that women in the follicular phase exhibit higher coagulability. As LBNP has been used previously to assess coagulation changes in whole blood samples in men and women, effects of the menstrual phase on coagulation was assessed during LBNP.

## 2. Experimental Section

An approval was obtained from the Ethics Committee of the Medical University of Graz before starting the study. Subjects also provided written and informed consent.

### 2.1. Subjects

The selection criteria included females of ages 18–35 years and heights 160–180 cm. The criteria for exclusion were endurance athletes, smokers, pregnant subjects, and subjects with/having a family history of cardiovascular diseases, thrombosis, coagulation disorders, orthostatic intolerance, or intake of any medication known to affect the coagulation system. Women on oral contraceptive pills (OCP) were not excluded. Each woman was asked about their menstrual cycle. The two phases were identified based on the following: early follicular phase (3–11 days after the commencement of the periods) and mid-luteal phase (21–27 days after commencement of the periods).

### 2.2. Sample Size

The number of participants required to show statistical significance were based on previously published studies, in which coagulation parameters were assessed during central hypovolemia [4,12,13]. The sample size calculation utilized the effects of LBNP on coagulation tests and markers of thrombin formation. An error probability (α) of 0.05, as well as the power (1- β) of 0.80 and an average effect size (d) of 0.5, was used to calculate the sample size [12]. The sample size calculation showed that *n* = 7 would be adequate to establish statistical significance. This was also the sample size that was used to assess effects of LBNP-induced hypovolemia on coagulation changes in males [14].

### 2.3. Study and LBNP Protocol

Prior to the experiments, the subjects were advised to refrain from coffee and alcohol for up to 24 h. The testing sessions were performed at 09:00–11:00. in a partially darkened quiet room. Room temperature was maintained at 23–24 °C and humidity at 55–60%.

Each experiment commenced with the subject in a supine position for 30 min, during which time the electrodes for cardiovascular monitoring were placed on the subjects [15]. A graded LBNP protocol was used: suction started at −10 mmHg and then was increased in 5 min intervals by −10 mmHg, until a maximum suction of −40 mmHg (a level at which the fluid shifts and central hypovolemia are similar to upright standing and most subjects are able to tolerate without developing presyncopal signs and symptoms) [16,17]. Finally, a 10 min recovery period was allowed.

### 2.4. Blood Sampling

Blood sampling occurred from a vein in the antecubital fossa at baseline, end of −40 mmHg LBNP, and at 10 min post-LBNP. Blood was collected into pre-citrated Vacuette^®^ containers (Greiner Bio-one GmbH, Kremsmünster, Austria), and within 3 h of sampling, the following were analyzed: TEM, WB, aggregation of platelets). A PPP sample was then obtained by centrifuging a part of the blood sample at room temperature at 1200 g for 15 min. PPP was used to measure conventional clotting times, pro- and anticoagulant proteins, thrombin generation curves, and markers of thrombin generation. Blood cell counts, Hematocrit, Hemoglobin, mean corpuscular hemoglobin concentration (MCHC), MCH, and mean corpuscular volume (MCV) were determined using an Automated Hematology Analyzer.

Standard coagulation tests and hematocrit and blood cell counts: Activated partial thromboplastin time (aPTT), prothrombin time (PT), plasma levels of protein C (PC), protein S (PS), coagulation factors FII, FVII, and FVIII were measured using BM/Hitachi 917, Austria). Blood cell counts, Hematocrit, Hemoglobin, MCHC, MCH, and MCV were determined using a Sysmex KX-21 N Automated Hematology Analyzer (Sysmex, IL, USA).

### 2.5. Thrombin Generation Markers and Endothelial Activation Markers

Prothrombin fragments 1 + 2 (F 1 + 2) and plasma thrombin-antithrombin (TAT) complexes were measured by ELISA (Behring Diagnostics GmbH, Germany). Tissue-Plasminogen Activator (tPA) concentration was assessed via IMUBIND t-PA ELISA assay (American Diagnostica Germany), and tissue factor (TF) was measured using the ACTICHROME Tissue Factor ELISA kit (American Diagnostica, Germany).

### 2.6. Thrombin Generation Using Automated Fluorogenic Measurements

Calibrated automated thrombography (CAT) to monitor the Thrombin generation curves [18] was assessed using the Thrombinoscope BV, in The Netherlands. Plasma thrombin generation was measured via the following parameters: Lag Time: lag time preceding the thrombin burst, endogenous thrombin potential (ETP), Peak: peak height, time to peak (ttPeak), peak rate of thrombin generation (peak thrombin/(peak time–lag time)) (VELINDEX), and StartTail: the time point at which free thrombin disappears. To detect thrombin formation, these measurements were carried out in the presence of low amounts of tissue factor (TF).

### 2.7. Tissue Factor Triggered TEM Assay (Thrombelastometer) Provide the Following

Clot formation time (CFT): beginning of clot formation until the amplitude of thrombelastogram reaches 20 mm time; coagulation time (CT): initiation of the test to the initial fibrin formation time; maximum clot firmness (MCF): maximum strength of the final clot (in mm); and Alpha angle: represents the acceleration kinetics of fibrin build up and cross-linking [9] and is the angle between the line in the middle of the TEM graph and the line tangential to the developing “body” of the TEM graph.

### 2.8. Statistics

The GraphPad 7.0 Prism package was used to perform all statistical analyses. The participants’ anthropometric measurements are presented as means ± SD, while the other data are presented as means ± SEM. As the main objective of the study was to assess whether there is a difference in women across two phases of their menstrual cycle in their susceptibility to react to LBNP, differences between the three periods (supine rest, LBNP, and 10 min post-LBNP) were determined by means of repeated measures using ANOVA, which included the factors menstrual phase and interaction between menstrual phase and time. Dunn’s post-test was then used. Menstrual phase differences in basal coagulation values were analyzed using the Mann–Whitney U-test. Statistical significance was set at *p* < 0.05. *… *p* ≤ 0.05, **… *p* ≤ 0.01, ***… *p* ≤ 0.001.

## 3. Results

Seven women (26.0 ± 2.0 years, 62 ± 11 kg body weight, 1.67 ± 0.05 m height, and 22.0 ± 3.0 kg/m^2^ BMI) participated in this study. All participants completed the study protocol (that is, no presyncopal event occurred during LBNP application).

### 3.1. LBNP Effects on Standard Coagulation Parameters

aPTT baseline values were the same in early follicular (EF) and mid-luteal (ML; *p* = 0.125) phases. LBNP suction caused a similar significant shortening of aPTT in both phases (Table 1). PT baseline levels were significantly higher in ML than in EF (Figure 1, panel A). LBNP suction had no effect on PT values in both phases. Consequently, during LBNP and recovery, the PT values remained significantly higher in ML, as compared to EF.

The respective baseline levels of FII, FVII, FVIII, PC, and PS were the same in the two phases. LBNP suction caused a similar significant increase of FVIII levels in EF and ML, (Table 1). However, FVII, PC, and PS levels were not affected by LBNP.

### 3.2. LBNP Effects on Blood Cell Count, Hematocrit, Hemoglobin, MCHC, MCH, and MCV

The respective baseline level white blood cell (WBC) counts were the same in the two phases of the menstrual cycle. LBNP suction led to significant decreases in WBC in both phases, but post-LBNP leucocytosis was observed (Table 1). The respective baseline values of RBC, Hct, Hb, MCH, and platelet count (Plt) were the same in EF and ML. Baseline values of MCHC were significantly lower (Figure 2, panel A), and baseline values of MCV (Figure 2, panel B) were significantly higher in EF compared to ML. MCHC and MCH were affected by LBNP only in EF, while Hct, Hb, and RBC were affected by LBNP only in ML.

### 3.3. LBNP Effects on Thrombin Generation Markers

The respective baseline levels of F1 + 2 and TAT were the same in EF and ML. LBNP suction caused a similar significant increase of both F1 + 2 and TAT levels in both phases (Table 1).

### 3.4. Effects of LBNP on Endothelial Activation Markers

Baseline TF and tPA levels were the same in EF and ML and were not significantly altered by LBNP.

### 3.5. Thrombin Generation Using CAT

The respective baseline levels of the CAT values were the same in EF and ML and were not affected by LBNP in both phases. However, a notable exception was the ttPeak response. The ttPeak was significantly shorter in the ML phase compared to the EF phase during LBNP application (Figure 1, panel B).

### 3.6. Effects of LBNP on TEM Values

The respective baseline levels of the TEM values CT, CFT, MCF, and alpha were the same in EF and ML. LBNP caused a similar increase in alpha in both EF and ML; CT, CFT, and MCF were not affected by LBNP (Table 1).

## 4. Discussion

Our study indicates that applying LBNP leads to coagulation activation in both plasma and whole blood samples. During both phases of the menstrual cycle, women are susceptible to LBNP, as reflected in their decreased PTT and elevated FVIII, F1 + 2, and TAT levels (Table 1). Additionally, during the ML phase, the significantly higher prothrombin time values and shorter ttpeak values (implying earlier start of thrombin generation) suggest that women in the ML phase are relatively hypercoagulable compared to the early follicular phase (Figure 1). These results suggest that thrombosis occurs more during the mid-luteal phase, an observation that has not been reported previously but could have significant clinical application.

We recently reported that sex differences exist in coagulation in both resting conditions and during LBNP application [4]. Specifically, women are relatively hypercoagulable compared to men during resting baseline conditions, but men show a more pronounced LBNP-induced shift towards hypercoagulability [4]. The current study extends the above literature with the finding that respective baseline levels of FII, FVII, FVIII, PC, and PS are the same in the two phases, thus implying that the phase of the menstrual cycle does not affect coagulability during resting conditions. This is in agreement with what has been reported in most studies. For instance, a systematic review of hemostatic variables during the menstrual cycle concluded that there are no cyclic variations in FVIII [19]. During LBNP, significant increases of FVIII levels in EF and ML were seen (Table 1). This finding is in agreement with LBNP induced increases of FVIII that we previously reported in men and women [4]. FVII, PC, and PS levels were, however, not affected by LBNP. While FVIII plasma levels have been shown to be increased significantly by LBNP in men [7,20], as well as in women [4]. No study has reported that FVIII levels change with LBNP application during different phases of the menstrual cycle.

Baseline levels of RBC, Hct, Hb, mean corpuscular hemoglobin (MCH), and platelets were similar in both menstrual phases. Interestingly, Hct, Hb, and RBC were significantly lower during LBNP in the ML phase but not during the EF phase. It has been previously reported that the increases of RBC, Hb, and Hct are mainly attributable to the hemoconcentration, which occurs as fluid re-allocates into the extravascular space of the legs during LBNP [6,15,21,22]. However, it is possible that the progesterone-induced increases in plasma volume during hypotension, which has been reported from animal studies [23] and has been confirmed in our blood samples (higher plasma volume calculated by changes in Hct) [24], and higher progesterone in the ML phase could have led to the decreases in RBC, Hb, and hematocrit during the LBNP-induced central hypovolemia. It is possible that the observed changes in blood counts seen during hypovolemia in the mid-luteal phase could have contributed to the coagulation changes seen in this phase. However, we are not able to confirm this association. Similarly, another study, in which extracellular homeostasis during the menstrual phase in humans was investigated [25], reported that, in the luteal phase, the concentration of water in the serum is higher, thus leading these investigators to suggest that these changes during the luteal phase “…in a lower scale, mimic well=known changes that occur during pregnancy”. However, further research shows that differences in vascular permeability across phases of the menstrual cycle during hypotension still needs to be carried out.

Baseline values of mean corpuscular hemoglobin concentration (MCHC) were significantly lower (Figure 2, panel A), and baseline values of mean corpuscular volume (MCV) (Figure 2, panel B) were significantly higher in EF compared with ML. The changes in MCHC are in contradiction to what has been reported previously [26]. On the other hand, LBNP affected MCHC and MCH only during the EF phase (Table 1).

Our results show that, just like in men and women, the phases of the menstrual cycle do not affect platelet count during LBNP. While some studies have reported elevations in platelets following higher LBNP applications, which might be associated with pronounced hemoconcentration and splenic release of platelets [6,27,28], we did not see such changes across the menstrual phases, which could potentially be attributed to the lower suction level (−40 mmHg) used in our study.

The respective baseline level WBC counts were the same in the two phases of the menstrual cycle. Previously, similar levels of WBC have been reported in both sexes at baseline [29] LBNP suction, which led to similar significant decreases in WBC in both phases (Table 1). In contradiction with the results of van Helmond et al., who observed a significant LBNP-induced leucocytosis in both men and women [2], we observed significant reduction in WBC counts during LBNP in both phases. However, post-LBNP leucocytosis was seen. The lower level of suction applied in our study (−40 mmHg vs. −45 mmHg applied in van Helmond et al. [2]) and the varying duration of LBNP application (20 min vs. 15 min applied in van Helmond et al., respectively [2]) might be responsible for the delayed leucocytosis that we observed.

The respective baseline levels of plasma markers of thrombin formation, F1 + 2, and TAT were the same in EF and ML. LBNP suction caused a similar significant increase of both F1 + 2 and TAT levels in both phases (Table 1).

Baseline levels of endothelial activation markers TF and tPA levels were the same in EF and ML and were not significantly altered by LBNP. Thus, LBNP appears to be a suitable method not only to simulate central hypovolaemia but also to expose individuals to a procoagulant challenge without massive endothelial activation associated with a possible risk to induce subsequent thrombosis in varying phases of the menstrual cycle. On the other hand, sit-to-stand tests or LBNP in combination with graded LBNP could potentially activate the endothelium and coagulation cascade with direct consequences [14].

The respective baseline levels of the CAT values were the same in EF and ML and were not affected by LBNP in either phase. However, a notable exception was the ttPeak response, which was significantly shorter in the ML phase compared to the EF phase during LBNP application (Figure 1, panel B). Baseline CAT values thus indicate similar coagulability across the phases. LBNP suction induced shorter ttPeaks in the ML phase (which indicate a faster maximum thrombin formation), thus implying a hypercoagulable state compared with the baseline.

The respective baseline levels of the TEM values CT, CFT, MCF, and alpha were the same in EF and ML. LBNP caused a similar increase in alpha in both EF and ML; CT, CFT, and MCF were not affected by LBNP (Table 1). TEM measurements appear to indicate a similar susceptibility of women to react on LBNP suction across the menstrual phases.

A possible limitation of this study is that we used the number of days after the start of the menstrual phase to calculate the phase [30]. While this may have led to some inaccuracies, we tried to minimize this by including only women with normal periods and/or not having cycles longer than 28 days and by measuring estrogen and progesterone levels at each phase of the menstrual cycle. We observed that estradiol levels were 62.84 ± 26.49 pg/mL (*n* = 6) in the early follicular phase and. 36.51 ± 12.84 pg/mL (*n* = 7) in the mid-luteal phase. Progesterone levels were 1.46 ± 0.5 ng/mL (*n* = 6) in the early follicular phase and 3.35 ± 3.47 ng/mL (*n* = 7) in the mid-luteal phase. Another limitation is that the number of women taking oral contraceptives (OCs) in our study was not controlled, as OCP intake was not an exclusion criterion. Our subject population (*n* = 7) showed the following characteristics: Women taking OCs: *n* = 3; women not taking OCs: *n* = 2; women not disclosing OC information: *n* = 2. However, we do not think that this an important limitation of our study, as assessment of the effects of OCs on coagulation was not the aim of the study. Moreover, the usage of OCs has been suggested to have only partial effects on the coagulation system. Lowe et al. (1997) have shown that only two coagulation-active proteins are influenced by OCs [10]. They reported that OC users have greater coagulation factor IX plasma levels and lower plasma PS values, both of which can potentially tip the coagulation cascade towards hypercoagulability [10]. However, no significant increases in coagulation activation was actually found in that study. As the effects of OCs on coagulation is still not well established, we suggest that future studies should be carried out that examine the role of OCP in coagulation.

Furthermore, our sample size may appear to be rather small (*n* = 7). However, sample size calculations carried out for this study revealed that this is an adequate number. Moreover, the strength of this study is that the same participants were tested two times (once in the follicular phase and once in the mid-luteal phase). Most of the other studies utilized greater number of females but did not test the same female in each phase. Finally, we have previously used a similar number of participants to observe coagulation changes during hypovolemia [14].

Another limitation of this study is that the effects of estrogen on coagulation factor XII were not investigated. Future studies should determine the influence of estrogens on the hemostatic system, especially as estrogens increase FXII and decrease platelet reactivity.

## 5. Conclusions

In conclusion, our study shows that:(1)During both phases, women are susceptible to increased coagulation during LBNP, as reflected in their decreased PTT and elevated FVIII, F1 + 2, and TAT levels;(2)During the mid-luteal phase, greater prothrombin time and shorter ttpeak values (implying faster maximum thrombin formation) suggest that women in the mid-luteal phase are relatively hypercoagulable compared with the early follicular phase;(3)LBNP represents a mild but efficient stimulus to expose individuals to a procoagulant challenge. It has been suggested recently that a simple sit-to-stand test might also have the potential to identify individuals with an increased risk of thrombosis [31]. However, a stand test is accompanied by an activation of both the endothelium and the coagulation cascade and might therefore not be appropriate in patients with higher risk of thrombosis. The present study shows that LBNP is a very mild but efficient coagulation stimulus and, thus, might be a suitable tool to screen for thrombosis.

## Figures and Tables

**Figure 1 jcm-09-03118-f001:**
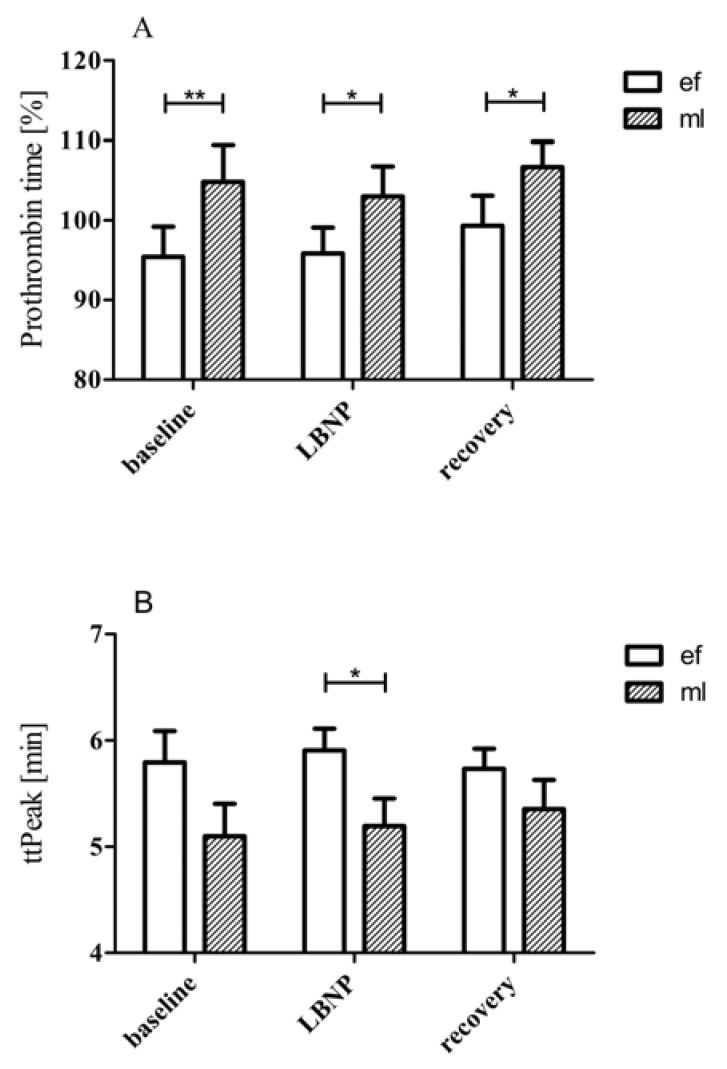
Prothrombin time and ttpeak values of women in early follicular and mid-luteal phases. During the ML phase, the significantly higher prothrombin time values and shorter ttpeak values (implying earlier start of thrombin generation) suggest that women in the ML phase are relatively hypercoagulable compared to those in the early follicular phase. Data are shown as mean ± SD. *… *p* ≤ 0.05, **… *p* ≤ 0.01.

**Figure 2 jcm-09-03118-f002:**
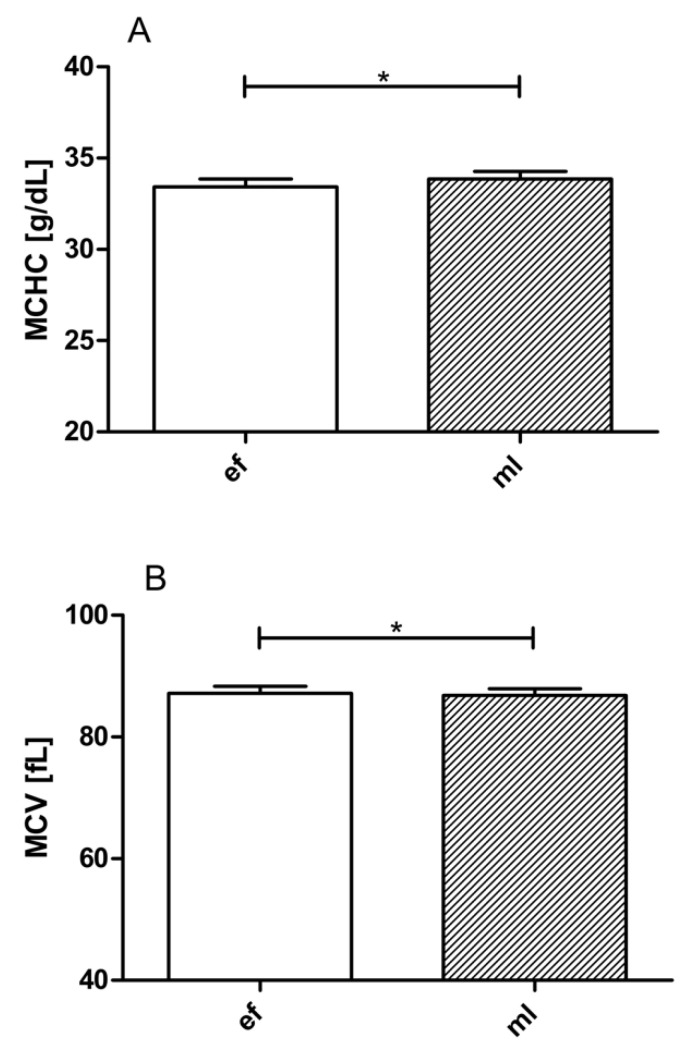
Baseline levels of mean corpuscular hemoglobin concentration (MCHC) values (panel A) and mean corpuscular volume (MCV) (panel B) during EF and ML phases. Data are expressed as means ± SEM. *… *p* ≤ 0.05.

**Table 1 jcm-09-03118-t001:** An overview of all the coagulation and hematological values at rest, during lower body negative pressure (LBNP) and recovery. Significant changes are highlighted in bold.

		Baseline	LBNP	Recovery	*P*_time_ (ANOVA)
**Early follicular phase**	aPTT [s]	36.14 ± 0.94	34.37 ± 0.96	33.13 ± 1.41	**0.011**
	PT [%]	97.00 ± 4.08	95.83 v 3.28	99.33 ± 3.77	0.281
	F II [%]	87.00 ± 7.96	87.33 ± 8.09	91.83 ± 5.04	0.445
	FVII [%]	104.70 ± 7.67	106.00 ± 7.27	104.60 ± 6.92	0.861
	F VIII [%]	71.57 ± 7.53	79.71 ± 8.09	90.29 ± 10.16	**0.003**
	PC [%]	96.67 ± 5.73	101.50 ± 8.57	101.00 ± 5.84	0.605
	PS [%]	91.83 ± 15.99	84.83 ± 11.90	76.67 ± 4.43	0.403
	WBC [10^3^/µL]	4.61 ± 0.50	4.11 ± 0.42	4.73 ± 0.61	**0.009**
	Plt [10^3^/mL]	218.30 ± 19.88	218.40 ± 20.20	224.90 ± 22.41	0.257
	RBC [10^6^/mL]	3.96 ± 0.11	3.98 ± 0.12	4.053 ± 0.13	0.170
	Hct [%]	34.49 ± 0.52	34.60 ± 0.68	35.23 ± 0.73	0.182
	Hb [g/dL]	11.54 ± 0.27	11.67 ± 0.35	11.90 ± 0.37	0.055
	MCHC [g/dL]	33.43 ± 0.43	33.70 ± 0.43	33.73 ± 0.43	**0.004**
	MCH [pg]	29.14 ± 0.39	29.36 ± 0.39	29.39 ± 0.42	**0.038**
	MCV [fL]	87.19 ± 1.12	87.20 ± 1.13	87.10 ± 1.12	0.733
	F1 + 2 [pmol/L]	225.3 ± 17.20	415.5 ± 71.37	731.3 ± 154.4	**0.002**
	TAT [ng/mL]	5.25 ± 1.66	29.81 ± 8.51	49.54 ± 8.71	**0.001**
	TF [pg/mL]	313.30 ± 27.39	362.10 ± 37.51	576.90 ± 170.30	0.121
	tPA [ng/mL]	1.58 ± 0.50	1.35 ± 0.41	1.39 ± 0.40	0.397
	LT [min]	2.43 ± 0.12	2.39 ± 0.06	2.30 ± 0.08	0.321
	ETP [nM•min]	1464 ± 203	1406 ± 186	1421 ± 214	0.137
	Peak [nM]	250.30 ± 36.66	232.80 ± 33.72	241.00 ± 35.11	0.081
	ttPeak [min]	5.79 ± 0.30	5.91 ± 0.21	5.73 ± 0.19	0.485
	VelIndex [nm/min]	81.33 ± 18.66	70.01 ± 13.86	73.86 ± 14.40	0.126
	StartTail [min]	21.87 ± 0.39	21.98 ± 0.27	22.00 ± 0.38	0.937
	CT [s]	167.00 ± 17.02	159.60 ± 15.17	163.40 ± 15.66	0.391
	CFT [s]	142.30 ± 20.35	115.90 ± 12.59	117.70 ± 12.47	0.052
	MCF [mm]	58.14 ± 2.22	59.43 ± 1.96	59.57 ± 1.88	0.205
	Alpha [°]	63.71 ± 3.19	67.57 ± 2.25	67.00 ± 2.23	**0.044**
**Mid-luteal phase**	aPTT [s]	34.70 ± 1.35	33.17 ± 1.69	32.21 ± 1.54	0.009
	PT [%]	104.9 ± 4.63	103.0 ± 3.77	106.07 ± 3.14	0.104
	F II [%]	92.33 ± 6.44	90.67 ± 6.66	92.00 ± 6.18	0.750
	FVII [%]	106.5 ± 13.6	107.7 ± 15.1	108.0 ± 13.7	0.849
	F VIII [%]	87.20 ± 13.32	94.00 ± 15.09	96.00 ± 12.59	**0.024**
	PC [%]	88.25 ± 16.06	96.67 ± 3.00	111.00 ± 7.27	0.312
	PS [%]	67.86 ± 10.41	72.86 ± 8.23	76.43 ± 7.24	0.328
	WBC [10^3^/µL]	4.64 ± 0.43	4.07 ± 0.37	4.90 ± 0.47	**0.001**
	Plt [10^3^/mL]	225.00 ± 18.63	218.40 ± 16.07	223.60 ± 18.40	0.430
	RBC [10^6^/mL]	3.91 ± 0.13	3.89 ± 0.15	4.02 ± 0.15	**0.012**
	Hct [%]	33.84 ± 0.87	33.70 ± 1.05	34.81 ± 1.00	**0.011**
	Hb [g/dL]	11.47 ± 0.35	11.43 ± 0.42	11.81 ± 0.42	**0.014**
	MCHC [g/dL]	33.86 ± 0.41	33.89 ± 0.45	33.90 ± 0.44	0.933
	MCH [pg]	29.39 ± 0.47	29.43 ± 0.48	29.41 ± 0.48	0.946
	MCV [fL]	86.80 ± 1.11	86.86 ± 1.17	86.79 ± 1.14	0.902
	F1 + 2 [pmol/L]	212.3 ± 27.80	546.2 ± 99.37	584.3 ± 108.7	**0.036**
	TAT [ng/mL]	3.30 ± 0.32	41.77 ± 8.81	41.19 ± 9.96	**0.004**
	TF [pg/mL]	295.00 ± 64.93	345.50 ± 83.44	547.50 ± 198.60	0.330
	tPA [ng/mL]	1.96 ± 0.57	2.40 ± 0.73	3.24 ± 0.75	0.287
	LT [min]	2.13 ± 0.10	2.22 ± 0.11	2.27 ± 0.14	0.184
	ETP [nM•min]	1688 ± 214	1645 ± 210	1645 ± 197	0.347
	Peak [nM]	312.80 ± 43.15	304.50 ± 42.68	297.30 ± 38.44	0.315
	ttPeak [min]	5.10 ± 0.31	5.19 ± 0.26	5.35 ± 0.28	0.239
	VelIndex [nm/min]	116.60 ± 24.64	110.90 ± 21.30	103.20 ± 19.57	0.079
	StartTail [min]	21.46 ± 0.43	21.55 ± 0.39	21.44 ± 0.41	0.906
	CT [s]	155.60 ± 13.11	144.40 ± 13.00	148.40 ± 13.48	0.076
	CFT [s]	121.70 ± 15.53	107.30 ± 20.56	99.86 ± 11.04	0.122
	MCF [mm]	60.86 ± 2.06	62.00 ± 1.98	62.00 ± 2.01	0.500
	Alpha [°]	66.14 ± 2.49	69.57 ± 3.12	70.00 ± 2.05	**0.035**

Legend: calibrated automated thrombography (CAT); clot formation time (CFT); coagulation factor (F II, VII, VIII); coagulation time (CT); endogenous thrombin potential (ETP); prothrombin fragment 1 + 2 (F 1 + 2); hematocrit (Hct); hemoglobin (Hb); head-up tilt (HUT); lower body negative pressure (LBNP); Lag time (LT); maximum clot firmness (MCF); mean corpuscular hemoglobin (MCH); mean corpuscular hemoglobin concentration (MCHC); mean corpuscular volume (MCV); oral contraceptives (OCs); protein C (PC); platelet count (Plt); protein S (PS); prothrombin times (PT); platelet poor plasma (PPP); red blood cell (RBC) counts; standard deviation (SD); standard error of mean (SEM); thrombin-antithrombin (TAT) complex; thrombelastometry (TEM); tissue factor (TF); tissue-plasminogen activator (t-PA); time to peak (ttPeak); whole blood (WB); white blood cell (WBC) counts.

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
