# Peer review of "Menstrual Phase Affects Coagulation and Hematological Parameters during Central Hypovolemia"

_jcm, 2020, doi:10.3390/jcm9103118_

Round 1

Reviewer 1 Report

This is a prospective study about coagulation changes during hypovolemia depending on the menstrual phase of women. 

I appreciate science on women since hormonal depending diseases or influences are widely ignored in the scientific settings.

I have a few comments.

General:

I doubt that seven patients are really enough to receive valid results since one different measurement of one single woman would influence the results essentially. In the methods section you write about the sample size calculation but it is unclear what the result of this calculation was. Please add the result of the sample size calculation and please address this problem of low proband number more in your limiation section.

Further, please address the influence of the contact (intrinsic) pathway on the coagulatory status of the women. It is suggested that estrogen has an influence on the FXII. Have you measured also FXII and FXI in the course of your study? If so, please add these results into your manuscript. If not, please add this possible influence in your discussion section.

Minor:

Line 90: There is an empty bracket, maybe there is some information missing. please add.

Line 107: there are two "a" in the sentence.

Line 136: you have called the midline of TEM graph "middle of TEG tracing". This might be confusing to the readers since TEG is a different technology within the viscoelastic tests. Please stick with TEM and rename this phrase.

Author Response

General:

I doubt that seven patients are really enough to receive valid results since one different measurement of one single woman would influence the results essentially. In the methods section you write about the sample size calculation but it is unclear what the result of this calculation was. Please add the result of the sample size calculation and please address this problem of low proband number more in your limiation section.

We have now added the following text in the sample size calculations section:

“The sample size calculation utilized the effects of LBPN on coagulation tests and markers of thrombin formation. Furthermore, sample size calculation showed that n=7 would be adequate to establish statistical significance.  This was also the sample size that was used when we previously published coagulation changes in males [14].”

We have now added the following text in the Limitations:

“Furthermore, our sample size may appear to be rather small (n=7). However, sample size calculations carried out for this study revealed that this is an adequate number. Moreover, the strength of this study is that the same participants were tested two times (once in follicular phase and once in the mid-luteal phase). Most of the other studies utilized greater number of females but did not test the same female in each phase. Finally, we have previously used similar number of participants to observe coagulation changes during hypovolemia [14].”

Further, please address the influence of the contact (intrinsic) pathway on the coagulatory status of the women. It is suggested that estrogen has an influence on the FXII. Have you measured also FXII and FXI in the course of your study? If so, please add these results into your manuscript. If not, please add this possible influence in your discussion section.

No, we did not measure FXII and FX1 (due to the high costs associated with these measurements). We are aware that estrogens have several infuences, including increases in FXII levels, and decreases in platelet reactivity. So the influence of estrogens are manifold.

We have now added this in the limitations section:

“Another limitation of this study is that the effects of estogens oN FXII was not investigated. Future studies should determine the influence of estrogen on FXII and FX1, especially as estrogens increase FXII and decrease platelet reactivity.”

Minor:

Line 90: There is an empty bracket, maybe there is some information missing. please add. Thank you for your comment. We added the missing information: (α) 

Line 107: there are two "a" in the sentence.
We deleted the redundant “a”.

Line 136: you have called the midline of TEM graph "middle of TEG tracing". This might be confusing to the readers since TEG is a different technology within the viscoelastic tests. Please stick with TEM and rename this phrase.
Thank you for pointing this out. We renamed this phrase according to your comment: TEG tracing TMG graph

Reviewer 2 Report

The authors applied a testing modality, lower body negative pressure, to test the hypothesis that women demonstrate higher levels of coagulability during certain phases of the menstrual cycle. My sense is this paper provides an interesting perspective; however, I question the practicality of using LBNP in clinical practice and the generalizability of this data to patients with increased thrombosis risk.

Minor issues:

  1. The authors may consider un-bolding the text, "all blood count determinations" on page 2, line 54.
  2. The authors may want offer a discussion on how LBNP would be contextualized in general practice. If not, I would recommend deleting the final paragraph in the introduction section.
  3. The authors may want to put something in the parentheses on page 2, line 90.
  4. The authors may want to delete "across the phases," from page 4, line 155. ML and EF were the only phases studied and the authors have given that data.
  5. The authors briefly describe changes in common CBC parameters from progesterone on page 8, line 233, during hypotension. Do the authors mean to suggest or should it be suggested that these fluid shifts are related to the coagulability changes identified in their study?

Author Response

Minor issues:

  1. The authors may consider un-bolding the text, "all blood count determinations" on page 2, line 54.

 The text is now not in bold anymore.

  1. The authors may want offer a discussion on how LBNP would be contextualized in general practice. If not, I would recommend deleting the final paragraph in the introduction section.

LBNP is a research tool. Therefore, it is not routinely used in general practice. We have, therefore, deleted the final paragraph in the introduction section.

The results from our study have clinical implications. Patients with an elevated risk for thrombosis in different phases of the menstrual cycle might be identifiable. An anticoagulant treatment of these singled-out patients might be essential to prevent future thrombotic events, especially during certain phases of the menstrual cycle.

  1. The authors may want to put something in the parentheses on page 2, line 90.

We added the missing information in the parentheses: (α)

  1. The authors may want to delete "across the phases," from page 4, line 155. ML and EF were the only phases studied and the authors have given that data.

Thank you for your comment, we now deleted the following: across the phases

  1. The authors briefly describe changes in common CBC parameters from progesterone on page 8, line 233, during hypotension. Do the authors mean to suggest or should it be suggested that these fluid shifts are related to the coagulability changes identified in their study?

This section mainly aims to explain the changes in CBC parameters induced by LBNP induced hypovolemia. It would, however, be difficult to conclude/ suggest that the changes in CBC parameters (decreases in Hb, hematocrit) could be responsible for the chance in coagulability observed during this phase. While the CBC parameters in some way could have contributed to the hyper-coagulability seen during this phase, it is difficult to establish whether these changes in CBC actually are responsible for the increase coagulability during this phase.

To avoid unfounded assumptions, we have now placed in the following text:
“ It is possible that the observed changes in blood counts seen during hypovolemia in the mid-luteal phase could have contributed to the coagulation changes seen in this phase. However, we are not able to confirm this association.”

Round 2

Reviewer 2 Report

My sense is that the authors have done a fine job revising this manuscript. However, I think this paper would be more appropriate in a research journal, rather than the journal of clinical medicine. As the authors have shared, this study uses a research technique. I concede that someday we may have a way to assess & manage patients, but currently this paper doesn't have implications for clinical medicine.